# NEURAL SKETCH LEARNING FOR CONDITIONAL PROGRAM GENERATION

**Vijayaraghavan Murali, Letao Qi, Swarat Chaudhuri, and Chris Jermaine**
Department of Computer Science
Rice University
Houston, TX 77005, USA.
{vijay, letao.qi, swarat, cmj4}@rice.edu

## ABSTRACT

We study the problem of generating source code in a strongly typed, Java-like programming language, given a label (for example a set of API calls or types) carrying a small amount of information about the code that is desired. The generated programs are expected to respect a "realistic" relationship between programs and labels, as exemplified by a corpus of labeled programs available during training.

Two challenges in such *conditional program generation* are that the generated programs must satisfy a rich set of syntactic and semantic constraints, and that source code contains many low-level features that impede learning. We address these problems by training a neural generator not on code but on *program sketches*, or models of program syntax that abstract out names and operations that do not generalize across programs. During generation, we infer a posterior distribution over sketches, then concretize samples from this distribution into type-safe programs using combinatorial techniques. We implement our ideas in a system for generating API-heavy Java code, and show that it can often predict the entire body of a method given just a few API calls or data types that appear in the method.

## 1 INTRODUCTION

Neural networks have been successfully applied to many generative modeling tasks in the recent past (Oord et al., 2016; Ha & Eck, 2017; Vinyals et al., 2015). However, the use of these models in generating highly structured text remains relatively understudied. In this paper, we present a method, combining neural and combinatorial techniques, for the condition generation of an important category of such text: *the source code of programs in Java-like programming languages*.

The specific problem we consider is one of supervised learning. During training, we are given a set of programs, each program annotated with a label, which may contain information such as the set of API calls or the types used in the code. Our goal is to learn a function $g$ such that for a test case of the form $(X, \mathsf{Prog})$ (where $\mathsf{Prog}$ is a program and $X$ is a label), $g(X)$ is a compilable, type-safe program that is equivalent to $\mathsf{Prog}$.

This problem has immediate applications in helping humans solve programming tasks (Hindle et al., 2012; Raychev et al., 2014). In the usage scenario that we envision, a human programmer uses a label to specify a small amount of information about a program that they have in mind. Based on this information, our generator seeks to produce a program equivalent to the "target" program, thus performing a particularly powerful form of code completion.

Conditional program generation is a special case of *program synthesis* (Manna & Waldinger, 1971; Summers, 1977), the classic problem of generating a program given a constraint on its behavior. This problem has received significant interest in recent years (Alur et al., 2013; Gulwani et al., 2017). In particular, several neural approaches to program synthesis driven by *input-output examples* have emerged (Balog et al., 2017; Parisotto et al., 2016; Devlin et al., 2017). Fundamentally, these approaches are tasked with associating a program's syntax with its semantics. As doing so in general is extremely hard, these methods choose to only generate programs in highly controlled domain-specific languages. For example, Balog et al. (2017) consider a functional language in which the

only data types permitted are integers and integer arrays, control flow is linear, and there is a sum total of 15 library functions. Given a set of input-output examples, their method predicts a vector of binary attributes indicating the presence or absence of various tokens (library functions) in the target program, and uses this prediction to guide a combinatorial search for programs.

In contrast, in conditional program generation, we are already given a set of tokens (for example library functions or types) that appear in a program or its metadata. Thus, we sidestep the problem of learning the semantics of the programming language from data. We ask: does this simpler setting permit the generation of programs from a much richer, Java-like language, with one has thousands of data types and API methods, rich control flow and exception handling, and a strong type system?

While simpler than general program synthesis, this problem is still highly nontrivial. Perhaps the central issue is that to be acceptable to a compiler, a generated program must satisfy a rich set of structural and semantic constraints such as "do not use undeclared variables as arguments to a procedure call" or "only use API calls and variables in a type-safe way". Learning such constraints automatically from data is hard. Moreover, as this is also a supervised learning problem, the generated programs also have to follow the patterns in the data while satisfying these constraints.

We approach this problem with a combination of neural learning and type-guided combinatorial search (Feser et al., 2015). Our central idea is to learn not over source code, but over tree-structured syntactic models, or *sketches*, of programs. A sketch abstracts out low-level names and operations from a program, but retains information about the program's control structure, the orders in which it invokes API methods, and the types of arguments and return values of these methods. We propose a particular kind of probabilistic encoder-decoder, called a Gaussian Encoder-Decoder or GED, to learn a distribution over sketches conditioned on labels. During synthesis, we sample sketches from this distribution, then flesh out these samples into type-safe programs using a combinatorial method for program synthesis. Doing so effectively is possible because our sketches are designed to contain rich information about control flow and types.

We have implemented our approach in a system called BAYOU.[1] We evaluate BAYOU in the generation of API-manipulating Android methods, using a corpus of about 150,000 methods drawn from an online repository. Our experiments show that BAYOU can often generate complex method bodies, including methods implementing tasks not encountered during training, given a few tokens as input.

## 2 PROBLEM STATEMENT

Now we define conditional program generation. Assume a universe $\mathbb{P}$ of *programs* and a universe $\mathbb{X}$ of *labels*. Also assume a set of training examples of the form $\{(\mathsf{X}_1, \mathsf{Prog}_1), (\mathsf{X}_2, \mathsf{Prog}_2), ...\}$, where each $\mathsf{X}_i$ is a label and each $\mathsf{Prog}_i$ is a program. These examples are sampled from an unknown distribution $Q(X, Prog)$, where $X$ and $Prog$ range over labels and programs, respectively.[2]

We assume an equivalence relation $Eqv \subseteq \mathbb{P} \times \mathbb{P}$ over programs. If $(\mathsf{Prog}_1, \mathsf{Prog}_2) \in Eqv$, then $\mathsf{Prog}_1$ and $\mathsf{Prog}_2$ are *functionally equivalent*. The definition of functional equivalence differs across applications, but in general it asserts that two programs are "just as good as" one another.

The goal of *conditional program generation* is to use the training set to learn a function $g : \mathbb{X} \to \mathbb{P}$ such that the expected value $\mathbf{E}[I((g(X), Prog) \in Eqv)]$ is maximized. Here, $I$ is the indicator function, returning 1 if its boolean argument is true, and 0 otherwise. Informally, we are attempting to learn a function $g$ such that if we sample $(\mathsf{X}, \mathsf{Prog}) \sim Q(X, Prog)$, $g$ should be able to reconstitute a program that is functionally equivalent to $\mathsf{Prog}$, using only the label $\mathsf{X}$.

### 2.1 INSTANTIATION

In this paper, we consider a particular form of conditional program generation. We take the domain $\mathbb{P}$ to be the set of possible programs in a programming language called AML that captures the essence of API-heavy Java programs (see Appendix A for more details). AML includes complex control flow such as loops, `if-then` statements, and exceptions; access to Java API data types; and calls to Java API methods. AML is a strongly typed language, and by definition, $\mathbb{P}$ only includes programs

---

[1] BAYOU is publicly available at `https://github.com/capergroup/bayou`.
[2] We use italic fonts for random variables and sans serif — for example $\mathsf{X}$ — for values of these variables.

```
String s;                                              String s;
BufferedReader br;                                     BufferedReader br;
FileReader fr;                                          InputStreamReader isr;
try {                                                  try {
 fr = new FileReader($String);                          isr = new InputStreamReader($InputStream);
 br = new BufferedReader(fr);                            br = new BufferedReader(isr);
 while ((s = br.readLine()) != null) {}                 while ((s = br.readLine()) != null) {}
 br.close();                                           } catch (IOException _e) {
} catch (FileNotFoundException _e) {                   }
} catch (IOException _e) {
}
```

(a)                                                    (b)

Figure 1: Programs generated by BAYOU with the API method name `readLine` as a label. Names of variables of type `T` whose values are obtained from the environment are of the form `$T`.

that are type-safe.[3] To define labels, we assume three finite sets: a set $Calls$ of possible API calls in AML, a set $Types$ of possible object types, and a set $Keys$ of *keywords*, defined as words, such as "read" and "file", that often appear in textual descriptions of what programs do. The space of possible labels is $\mathbb{X} = 2^{Calls} \times 2^{Types} \times 2^{Keys}$ (here $2^S$ is the power set of $S$).

Defining $Eqv$ in practice is tricky. For example, a reasonable definition of $Eqv$ is that $(\mathsf{Prog}_1, \mathsf{Prog}_2) \in Eqv$ iff $\mathsf{Prog}_1$ and $\mathsf{Prog}_2$ produce the same outputs on all inputs. But given the richness of AML, the problem of determining whether two AML programs always produce the same output is undecidable. As such, in practice we can only measure success indirectly, by checking whether the programs use the same control structures, and whether they can produce the same API call sequences. We will discuss this issue more in Section 6.

## 2.2 EXAMPLE

Consider the label $\mathsf{X} = (\mathsf{X}_{Calls}, \mathsf{X}_{Types}, \mathsf{X}_{Keys})$ where $\mathsf{X}_{Calls} = \{\texttt{readLine}\}$ and $\mathsf{X}_{Types}$ and $\mathsf{X}_{Keys}$ are empty. Figure 1(a) shows a program that our best learner stochastically returns given this input. As we see, this program indeed reads lines from a file, whose name is given by a special variable `$String` that the code takes as input. It also handles exceptions and closes the reader, even though these actions were not directly specified.

Although the program in Figure 1-(a) matches the label well, failures do occur. Sometimes, the system generates a program as in Figure 1-(b), which uses an `InputStreamReader` rather than a `FileReader`. It is possible to rule out this program by adding to the label. Suppose we amend $\mathsf{X}_{Types}$ so that $\mathsf{X}_{Types} = \{\texttt{FileReader}\}$. BAYOU now tends to only generate programs that use `FileReader`. The variations then arise from different ways of handling exceptions and constructing `FileReader` objects (some programs use a `String` argument, while others use a `File` object). Figure 7 in the appendix shows two other top-five programs returned on this input.

## 3 TECHNICAL APPROACH

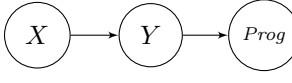

Figure 2: Bayes net for $Prog, X, Y$

Our approach is to learn $g$ via maximum conditional likelihood estimation (CLE). That is, given a distribution family $P(Prog|X, \theta)$ for a parameter set $\theta$, we choose $\theta^* = \arg\max_\theta \sum_i \log P(\mathsf{Prog}_i \mid \mathsf{X}_i, \theta)$. Then, $g(\mathsf{X}) = \arg\max_{\mathsf{Prog}} P(\mathsf{Prog}|\mathsf{X}, \theta^*)$.

The key innovation of our approach is that here, learning happens at a higher level of abstraction than $(\mathsf{X}_i, \mathsf{Prog}_i)$ pairs. In practice, Java-like programs contain many low-level details (for example, variable names and intermediate results) that can obscure patterns in code. Further, they contain complicated semantic rules (for example, for type safety) that are difficult to learn from data. In contrast, these are relatively easy for a combinatorial, syntax-guided program synthesizer (Alur et al., 2013) to deal with. However, synthesizers have a

---

[3]In research on programming languages, a program is typically judged as type-safe under a *type environment*, which sets up types for the program's input variables and return value. Here, we consider a program to be type-safe if it can be typed under *some* type environment.

notoriously difficult time figuring out the correct "shape" of a program (such as the placement of loops and conditionals), which we hypothesize should be relatively easy for a statistical learner.

Specifically, our approach learns over *sketches*: tree-structured data that capture key facets of program syntax. A sketch $Y$ does not contain low-level variable names and operations, but carries information about broadly shared facets of programs such as the types and API calls. During generation, a program synthesizer is used to generate programs from sketches produced by the learner.

Let the universe of all sketches be denoted by $\mathbb{Y}$. The sketch for a given program is computed by applying an *abstraction function* $\alpha : \mathbb{P} \to \mathbb{Y}$. We call a sketch $Y$ *satisfiable*, and write $sat(Y)$, if $\alpha^{-1}(Y) \neq \emptyset$. The process of generating (type-safe) programs given a satisfiable sketch $Y$ is probabilistic, and captured by a *concretization distribution* $P(Prog \mid Y, sat(Y))$. We require that for all programs Prog and sketches $Y$ such that $sat(Y)$, we have $P(\text{Prog} \mid Y) \neq 0$ only if $Y = \alpha(\text{Prog})$.

Importantly, the concretization distribution is fixed and chosen heuristically. The alternative of learning this distribution from source code poses difficulties: a single sketch can correspond to many programs that only differ in superficial details, and deciding which differences between programs are superficial and which are not requires knowledge about program semantics. In contrast, our heuristic approach utilizes known semantic properties of programming languages like ours — for example, that local variable names do not matter, and that some algebraic expressions are semantically equivalent. This knowledge allows us to limit the set of programs that we generate.

$$
\begin{array}{lll}
Y & ::= & \textbf{skip} \mid \textbf{call } \text{Cexp} \mid Y_1; Y_2 \mid \\
& & \textbf{if } \text{Cseq } \textbf{then } Y_1 \textbf{ else } Y_2 \mid \\
& & \textbf{while } \text{Cseq } \textbf{do } Y_1 \mid \textbf{try } Y_1 \text{ Catch} \\
\text{Cexp} & ::= & \tau_0.a(\tau_1, \ldots, \tau_k) \\
\text{Cseq} & ::= & \text{List of Cexp} \\
\text{Catch} & ::= & \textbf{catch}(\tau_1) Y_1 \ldots \textbf{catch}(\tau_k) Y_k
\end{array}
$$

Figure 3: Grammar for sketches

Let us define a random variable $Y = \alpha(Prog)$. We assume that the variables $X$, $Y$ and $Prog$ are related as in the Bayes net in Figure 2. Specifically, given $Y$, $Prog$ is conditionally independent of $X$. Further, let us assume a distribution family $P(Y|X, \theta)$ parameterized on $\theta$.

Let $Y_i = \alpha(\text{Prog}_i)$, and note that $P(\text{Prog}_i|Y) \neq 0$ only if $Y = Y_i$. Our problem now simplifies to *learning over sketches*, i.e., finding

$$
\begin{aligned}
\theta^* &= \arg\max_\theta \sum_i \log \sum_{Y:sat(Y)} P(\text{Prog}_i|Y)P(Y|X_i, \theta) \\
&= \arg\max_\theta \sum_i \log P(\text{Prog}_i|Y_i)P(Y_i|X_i, \theta) = \arg\max_\theta \sum_i \log P(Y_i|X_i, \theta). \quad (1)
\end{aligned}
$$

### 3.1 INSTANTIATION

Figure 3 shows the full grammar for sketches in our implementation. Here, $\tau_0, \tau_1, \ldots$ range over a finite set of *API data types* that AML programs can use. A data type, akin to a Java class, is identified with a finite set of *API method names* (including constructors), and $a$ ranges over these names. Note that sketches do not contain constants or variable names.

A full definition of the abstraction function for AML appears in Appendix B. As an example, API calls in AML have the syntax "$\textbf{call } e.a(e_1, \ldots, e_k)$", where $a$ is an API method, the expression $e$ evaluates to the object on which the method is called, and the expressions $e_1, \ldots, e_k$ evaluate to the arguments of the method call. We abstract this call into an *abstract method call* "$\textbf{call } \tau.a(\tau_1, \ldots, \tau_k)$", where $\tau$ is the type of $e$ and $\tau_i$ is the type of $e_i$. The keywords **skip**, **while**, **if-then-else**, and **try-catch** preserve information about control flow and exception handling. Boolean conditions Cseq are replaced by *abstract expressions*: lists whose elements abstract the API calls in Cseq.

## 4 LEARNING

Now we describe our learning approach. Equation 1 leaves us with the problem of computing $\arg\max_\theta \sum_i \log P(Y_i|X_i, \theta)$, when each $X_i$ is a label and $Y_i$ is a sketch. Our answer is to utilize an encoder-decoder and introduce a real vector-valued latent variable $Z$ to stochastically link labels and sketches: $P(Y|X, \theta) = \int_{Z \in \mathbb{R}^m} P(Z|X, \theta)P(Y|Z, \theta)dZ$.

$P(Y|Z, \theta)$ is realized as a probabilistic decoder mapping a vector-valued variable to a distribution over trees. We describe this decoder in Appendix C. As for $P(Z|X, \theta)$, this distribution can, in principle, be picked in any way we like. In practice, because both $P(Y|Z, \theta)$ and $P(Z|X, \theta)$ have neural components with numerous parameters, we wish this distribution to regularize the learner. To provide this regularization, we assume a Normal $(\vec{0}, \mathbf{I})$ prior on $Z$.

Recall that our labels are of the form $X = (X_{Calls}, X_{Types}, X_{Keys})$, where $X_{Calls}$, $X_{Types}$, and $X_{Keys}$ are sets. Assuming that the $j$-th elements $X_{Calls,j}$, $X_{Types,j}$, and $X_{Keys,j}$ of these sets are generated independently, and assuming a function $f$ for encoding these elements, let:

$$P(X|Z, \theta) = \left( \prod_j \text{Normal}(f(X_{Calls,j})|Z, \mathbf{I}\sigma^2_{Calls}) \right) \left( \prod_j \text{Normal}(f(X_{Types,j})|Z, \mathbf{I}\sigma^2_{Types}) \right)$$
$$\left( \prod_j \text{Normal}(f(X_{Keys,j})|Z, \mathbf{I}\sigma^2_{Keys}) \right).$$

That is, the encoded value of each $X_{Types,j}$, $X_{Calls,j}$ or $X_{Keys,j}$ is sampled from a high-dimensional Normal distribution centered at $Z$. If $f$ is 1-1 and onto with the set $\mathbb{R}^m$ then from Normal-Normal conjugacy, we have: $P(Z|X) = \text{Normal}\left( Z \mid \frac{\overline{X}}{1+n}, \frac{1}{1+n}\mathbf{I} \right)$, where

$$\overline{X} = \left( \sigma^{-2}_{Types} \sum_j f(X_{Types,j}) \right) + \left( \sigma^{-2}_{Calls} \sum_j f(X_{Calls,j}) \right) + \left( \sigma^{-2}_{Keys} \sum_j f(X_{Keys,j}) \right)$$

and $n = n_{Types}\sigma^{-2}_{Types} + n_{Calls}\sigma^{-2}_{Calls} + n_{Keys}\sigma^{-2}_{Keys}$. Here, $n_{Types}$ is the number of types supplied, and $n_{Calls}$ and $n_{Keys}$ are defined similarly.

Note that this particular $P(Z|X, \theta)$ only follows directly from the Normal $(\vec{0}, \mathbf{I})$ prior on $Z$ and Normal likelihood $P(X|Z, \theta)$ if the encoding function $f$ is 1-1 and onto. However, even if $f$ is *not* 1-1 and onto (as will be the case if $f$ is implemented with a standard feed-forward neural network) we can still use this probabilistic encoder, and in practice we still tend to see the benefits of the regularizing prior on $Z$, with $P(Z)$ distributed approximately according to a unit Normal. We call this type of encoder-decoder, with a single, Normally-distributed latent variable $Z$ linking the input and output, a Gaussian encoder-decoder, or GED for short.

Now that we have chosen $P(X|Z, \theta)$ and $P(Y|Z, \theta)$, we must choose $\theta$ to perform CLE. Note that:

$$\sum_i \log P(Y_i|X_i, \theta) = \sum_i \log \int_{Z \in \mathbb{R}^m} P(Z|X_i, \theta)P(Y_i|Z, \theta)dZ = \sum_i \log \mathbf{E}_{Z \sim P(Z|X_i, \theta)}[P(Y_i|Z, \theta)]$$
$$\geq \sum_i \mathbf{E}_{Z \sim P(Z|X_i, \theta)}[\log P(Y_i|Z, \theta)] = \mathcal{L}(\theta).$$

where the $\geq$ holds due to Jensen's inequality. Hence, $\mathcal{L}(\theta)$ serves as a lower bound on the log-likelihood, and so we can compute $\theta^* = \arg\max_\theta \mathcal{L}(\theta)$ as a proxy for the CLE. We maximize this lower bound using stochastic gradient ascent; as $P(Z|X_i, \theta)$ is Normal, we can use the reparameterization trick common in variational auto-encoders (Kingma & Welling, 2014) while doing so. The parameter set $\theta$ contains all of the parameters of the encoding function $f$ as well as $\sigma_{Types}$, $\sigma_{Calls}$, and $\sigma_{Keys}$, and the parameters used in the decoding distribution funciton $P(Y|Z, \theta)$.

## 5 COMBINATORIAL CONCRETIZATION

The final step in our algorithm is to "concretize" sketches into programs, following the distribution $P(Prog|Y)$. Our method of doing so is a type-directed, stochastic search procedure that builds on combinatorial methods for program synthesis (Schkufza et al., 2016; Feser et al., 2015).

Given a sketch $Y$, our procedure performs a random walk in a space of *partially concretized sketches* (PCSs). A PCS is a term obtained by replacing some of the abstract method calls and expressions in a sketch by AML method calls and AML expressions. For example, the term "$x_1.a(x_2); \tau_1.b(\tau_2)$",

which sequential composes an abstract method call to $b$ and a "concrete" method call to $a$, is a PCS. The state of the procedure at the $i$-th point of the walk is a PCS $\mathsf{H}_i$. The initial state is $\mathsf{Y}$.

Each state $\mathsf{H}$ has a set of *neighbors* $Next(\mathsf{H})$. This set consists of all PCS-s $\mathsf{H}'$ that are obtained by concretizing a single abstract method call or expression in $\mathsf{H}$, using variable names in a way that is consistent with the types of all API methods and declared variables in $\mathsf{H}$.

The $(i+1)$-th state in a walk is a sample from a predefined, heuristically chosen distribution $P(\mathsf{H}_{i+1} \mid \mathsf{H}_i,)$. The only requirement on this distribution is that it assigns nonzero probability to a state iff it belongs to $Next(\mathsf{H}_i)$. In practice, our implementation of this distribution prioritizes programs that are simpler. The random walk ends when it reaches a state $\mathsf{H}^*$ that has no neighbors. If $\mathsf{H}^*$ is fully concrete (that is, an AML program), then the walk is successful and $\mathsf{H}^*$ is returned as a sample. If not, the current walk is rejected, and a fresh walk is started from the initial state.

Recall that the concretization distribution $P(Prog|\mathsf{Y})$ is only defined for sketches $\mathsf{Y}$ that are satisfiable. Our concretization procedure does not *assume* that its input $\mathsf{Y}$ is satisfiable. However, if $\mathsf{Y}$ is not satisfiable, all random walks that it performs end with rejection, causing it to never terminate.

While the worst-case complexity of this procedure is exponential in the generated programs, it performs well in practice because of our chosen language of sketches. For instance, our search does not need to discover the high-level structure of programs. Also, sketches specify the types of method arguments and return values, and this significantly limits the search space.

## 6  EXPERIMENTS

Now we present an empirical evaluation of the effectiveness of our method. The experiments we describe utilize data from an online repository of about 1500 Android apps (and, 2017). We decompiled the APKs using JADX (Skylot, 2017) to generate their source code. Analyzing about 100 million lines of code that were generated, we extracted 150,000 methods that used Android APIs or the Java library. We then pre-processed all method bodies to translate the code from Java to AML, preserving names of relevant API calls and data types as well as the high-level control flow. Hereafter, when we say "program" we refer to an AML program.

|  | Min | Max | Median | Vocab |
|---|---|---|---|---|
| $\mathsf{X}_{Calls}$ | 1 | 9 | 2 | 2584 |
| $\mathsf{X}_{Types}$ | 1 | 15 | 3 | 1521 |
| $\mathsf{X}_{Keys}$ | 2 | 29 | 8 | 993 |
| $\mathsf{X}$ | 4 | 48 | 13 | 5098 |

Figure 4: Statistics on labels

From each program, we extracted the sets $\mathsf{X}_{Calls}$, $\mathsf{X}_{Types}$, and $\mathsf{X}_{Keys}$ as well as a sketch $\mathsf{Y}$. Lacking separate natural language dscriptions for programs, we defined keywords to be words obtained by splitting the names of the API types and calls that the program uses, based on camel case. For instance, the keywords obtained from the API call `readLine` are "read" and "line". As API method and types in Java tend to be carefully named, these words often contain rich information about what programs do. Figure 4 gives some statistics on the sizes of the labels in the data. From the extracted data, we randomly selected 10,000 programs to be in the testing and validation data each.

### 6.1  IMPLEMENTATION AND TRAINING

We implemented our approach in our tool called BAYOU, using TensorFlow (Abadi et al., 2015) to implement the GED neural model, and the Eclipse IDE for the abstraction from Java to the language of sketches and the combinatorial concretization.

In all our experiments we performed cross-validation through grid search and picked the best performing model. Our hyper-parameters for training the model are as follows. We used 64, 32 and 64 units in the encoder for API calls, types and keywords, respectively, and 128 units in the decoder. The latent space was 32-dimensional. We used a mini-batch size of 50, a learning rate of 0.0006 for the Adam gradient-descent optimizer (Kingma & Ba, 2014), and ran the training for 50 epochs.

The training was performed on an AWS "p2.xlarge" machine with an NVIDIA K80 GPU with 12GB GPU memory. As each sketch was broken down into a set of production paths, the total number of data points fed to the model was around 700,000 per epoch. Training took 10 hours to complete.

## 6.2 CLUSTERING

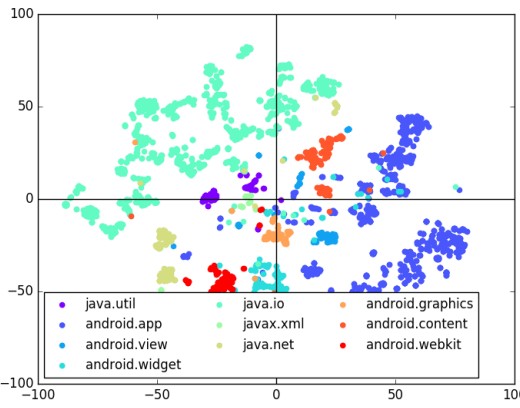

Figure 5: 2-dimensional projection of latent space

To visualize clustering in the 32-dimensional latent space, we provided labels X from the testing data and sampled Z from $P(Z|\mathsf{X})$, and then used it to sample a sketch from $P(Y|\mathsf{Z})$. We then used t-SNE (Maaten & Hinton, 2008) to reduce the dimensionality of Z to 2-dimensions, and labeled each point with the API used in the sketch Y. Figure 5 shows this 2-dimensional space, where each label has been coded with a different color. It is immediately apparent from the plot that the model has learned to cluster the latent space neatly according to different APIs. Some APIs such as `java.io` have several modes, and we noticed separately that each mode corresponds to different usage scenarios of the API, such as reading versus writing in this case.

## 6.3 ACCURACY

To evaluate prediction accuracy, we provided labels from the testing data to our model, sampled sketches from the distribution $P(Y|\mathsf{X})$ and concretized each sketch into an AML program using our combinatorial search. We then measured the number of test programs for which a program that is equivalent to the expected one appeared in the top-10 results from the model.

As there is no universal metric to measure program equivalence (in fact, it is an undecidable problem in general), we used several metrics to approximate the notion of equivalence. We defined the following metrics on the top-10 programs predicted by the model:

M1. This binary metric measures whether the expected program appeared in a syntactically equivalent form in the results. Of course, an impediment to measuring this is that the names of variables used in the expected and predicted programs may not match. It is neither reasonable nor useful for any model of code to learn the exact variable names in the training data. Therefore, in performing this equivalence check, we abstract away the variable names and compare the rest of the program's Abstract Syntax Tree (AST) instead.

M2. This metric measures the minimum *Jaccard distance* between the sets of sequences of API calls made by the expected and predicted programs. It is a measure of how close to the original program were we able to get in terms of sequences of API calls.

M3. Similar to metric M2, this metric measures the minimum Jaccard distance between the sets of API calls in the expected and predicted programs.

M4. This metric computes the minimum absolute difference between the number of statements in the expected and sampled programs, as a ratio of that in the former.

M5. Similar to metric M4, this metric computes the minumum absolute difference between the number of control structures in the expected and sampled programs, as a ratio of that in the former. Examples of control structures are branches, loops, and try-catch statements.

## 6.4 PARTIAL OBSERVABILITY

To evaluate our model's ability to predict programs given a small amount of information about its code, we varied the fraction of the set of API calls, types, and keywords provided as input from the testing data. We experimented with 75%, 50% and 25% observability in the testing data; the median number of items in a label in these cases were 9, 6, and 2, respectively.

| Model | Input Label Observability | | | |
|---|---|---|---|---|
| | 100% | 75% | 50% | 25% |
| GED-AML | 0.13 | 0.09 | 0.07 | 0.02 |
| GSNN-AML | 0.07 | 0.04 | 0.03 | 0.01 |
| GED-Sk | **0.59** | **0.51** | **0.44** | **0.21** |
| GSNN-Sk | 0.57 | 0.48 | 0.41 | 0.18 |

(a) M1. Proportion of test programs for which the expected AST appeared in the top-10 results.

| Model | Input Label Observability | | | |
|---|---|---|---|---|
| | 100% | 75% | 50% | 25% |
| GED-AML | 0.82 | 0.87 | 0.89 | 0.97 |
| GSNN-AML | 0.88 | 0.92 | 0.93 | 0.98 |
| GED-Sk | **0.34** | **0.43** | **0.50** | **0.76** |
| GSNN-Sk | 0.36 | 0.46 | 0.53 | 0.78 |

(b) M2. Average minimum Jaccard distance on the set of sequences of API methods called in the test program vs the top-10 results.

| Model | Input Label Observability | | | |
|---|---|---|---|---|
| | 100% | 75% | 50% | 25% |
| GED-AML | 0.52 | 0.58 | 0.61 | 0.77 |
| GSNN-AML | 0.59 | 0.64 | 0.68 | 0.83 |
| GED-Sk | **0.11** | **0.17** | **0.22** | **0.50** |
| GSNN-Sk | 0.13 | 0.19 | 0.25 | 0.52 |

(c) M3. Average minimum Jaccard distance on the set of API methods called in the test program vs the top-10 results.

| Model | Input Label Observability | | | |
|---|---|---|---|---|
| | 100% | 75% | 50% | 25% |
| GED-AML | 0.49 | 0.47 | 0.46 | 0.46 |
| GSNN-AML | 0.52 | 0.49 | 0.49 | 0.53 |
| GED-Sk | **0.05** | **0.06** | **0.06** | **0.09** |
| GSNN-Sk | **0.05** | **0.06** | **0.06** | **0.09** |

(d) M4. Average minimum difference between the number of statements in the test program vs the top-10 results.

| Model | Input Label Observability | | | |
|---|---|---|---|---|
| | 100% | 75% | 50% | 25% |
| GED-AML | 0.31 | 0.30 | 0.30 | 0.34 |
| GSNN-AML | 0.32 | 0.31 | 0.32 | 0.39 |
| GED-Sk | **0.03** | **0.03** | **0.03** | **0.04** |
| GSNN-Sk | **0.03** | **0.03** | **0.03** | **0.03** |

(e) M5. Average minimum difference between the number of control structures in the test program vs the top-10 results.

| Model | Metric | | | | |
|---|---|---|---|---|---|
| | M1 | M2 | M3 | M4 | M5 |
| GED-AML | 0.02 | 0.97 | 0.71 | 0.50 | 0.37 |
| GSNN-AML | 0.01 | 0.98 | 0.74 | 0.51 | 0.37 |
| GED-Sk | **0.23** | **0.70** | **0.30** | **0.08** | **0.04** |
| GSNN-Sk | 0.20 | 0.74 | 0.33 | **0.08** | **0.04** |

(f) Metrics for 50% obsevability evaluated only on unseen data

Figure 6: Accuracy of different models on testing data. GED-AML and GSNN-AML are baseline models trained over AML ASTs, GED-Sk and GSNN-Sk are models trained over sketches.

## 6.5 COMPETING MODELS

In order to compare our model with state-of-the-art conditional generative models, we implemented the Gaussian Stochastic Neural Network (GSNN) presented by (Sohn et al., 2015), using the same tree-structured decoder as the GED. There are two main differences: (i) the GSNN's decoder is also conditioned directly on the input label X in addition to Z, which we accomplish by concatenating its initial state with the encoding of X, (ii) the GSNN loss function has an additional KL-divergence term weighted by a hyper-parameter $\beta$. We subjected the GSNN to the same training and cross-validation process as our model. In the end, we selected a model that happened to have very similar hyper-parameters as ours, with $\beta = 0.001$.

## 6.6 EVALUATING SKETCHES

In order to evaluate the effect of sketch learning for program generation, we implemented and compared with a model that learns directly over programs. Specifically, the neural network structure is exactly the same as ours, except that instead of being trained on production paths in the sketches, the model is trained on production paths in the ASTs of the AML programs. We selected a model that had more units in the decoder (256) compared to our model (128), as the AML grammar is more complex than the grammar of sketches. We also implemented a similar GSNN model to train over AML ASTs directly.

Figure 6 shows the collated results of this evaluation, where each entry computes the average of the corresponding metric over the 10000 test programs. It takes our model about 8 seconds, on average, to generate and rank 10 programs.

When testing models that were trained on AML ASTs, namely the GED-AML and GSNN-AML models, we observed that out of a total of 87,486 AML ASTs sampled from the two models, 2525 (or 3%) ASTs were not even well-formed, i.e., they would not pass a parser, and hence had to be discarded from the metrics. This number is 0 for the GED-Sk and GSNN-Sk models, meaning that all AML ASTs that were obtained by concretizing sketches were well-formed.

In general, one can observe that the GED-Sk model performs best overall, with GSNN-Sk a reasonable alternative. We hypothesize that the reason GED-Sk performs slightly better is the regularizing prior on $Z$; since the GSNN has a direct link from $X$ to $Y$, it can choose to ignore this regularization. We would classify both these models as suitable for conditional program generation. However, the other two models GED-AML and GSNN-AML perform quite worse, showing that sketch learning is key in addressing the problem of conditional program generation.

## 6.7 GENERALIZATION

To evaluate how well our model generalizes to unseen data, we gather a subset of the testing data whose data points, consisting of label-sketch pairs $(X, Y)$, never occurred in the training data. We then evaluate the same metrics in Figure 6(a)-(e), but due to space reasons we focus on the 50% observability column. Figure 6(f) shows the results of this evaluation on the subset of 5126 (out of 10000) unseen test data points. The metrics exhibit a similar trend, showing that the models based on sketch learning are able to generalize much better than the baseline models, and that the GED-Sk model performs the best.

## 7 RELATED WORK

*Unconditional*, corpus-driven generation of programs has been studied before (Maddison & Tarlow, 2014; Allamanis & Sutton, 2014; Bielik et al., 2016), as has the generation of code snippets conditioned on a *context* into which the snippet is merged (Nguyen et al., 2013; Raychev et al., 2014; Nguyen & Nguyen, 2015). These prior efforts often use models like $n$-grams (Nguyen et al., 2013) and recurrent neural networks (Raychev et al., 2014) that are primarily suited to the generation of straight-line programs; almost universally, they cannot guarantee semantic properties of generated programs. Among prominent exceptions, Maddison & Tarlow (2014) use log-bilinear tree-traversal models, a class of probabilistic pushdown automata, for program generation. Bielik et al. (2016) study a generalization of probabilistic grammars known as probabilistic higher-order grammars. Like our work, these papers address the generation of programs that satisfy rich constraints such as the type-safe use of names. In principle, one could replace our decoder and the combinatorial concretizer, which together form an unconditional program generator, with one of these models. However, given our experiments, doing so is unlikely to lead to good performance in the end-to-end problem of conditional program generation.

There is a line of existing work considering the generation of programs from text (Yin & Neubig, 2017; Ling et al., 2016; Rabinovich et al., 2017). These papers use decoders similar to the one used in BAYOU, and since they are solving the text-to-code problem, they utilize attention mechanisms not found in BAYOU. Those attention mechanisms could be particularly useful were BAYOU extended to handle natural language evidence. The fundamental difference between these works and BAYOU, however, is the level of abstraction at which learning takes place. These papers attempt to translate text directly into code, whereas BAYOU uses neural methods to produce higher-level sketches that are translated into program code using symbolic methods. This two-step code generation process is central to BAYOU. It ensures key semantic properties of the generated code (such as type safety) and by abstracting away from the learner many lower-level details, it may make learning easier. We have given experimental evidence that this approach can give better results than translating directly into code.

Kusner et al. (2017) propose a variational autoencoder for context-free grammars. As an autoencoder, this model is generative, but it is not a conditional model such as ours. In their application of synthesizing molecular structures, given a particular molecular structure, their model can be used to search the latent space for similar valid structures. In our setting, however, we are not given a sketch but only a label for the sketch, and our task is learn a conditional model that can predict a whole sketch given a label.

Conditional program generation is closely related to *program synthesis* (Gulwani et al., 2017), the problem of producing programs that satisfy a given semantic specification. The programming language community has studied this problem thoroughly using the tools of combinatorial search and symbolic reasoning (Alur et al., 2013; Solar-Lezama et al., 2006; Gulwani, 2011; Feser et al., 2015). A common tactic in this literature is to put syntactic limitations on the space of feasible

programs (Alur et al., 2013). This is done either by adding a human-provided sketch to a problem instance (Solar-Lezama et al., 2006), or by restricting synthesis to a narrow DSL (Gulwani, 2011; Polozov & Gulwani, 2015).

A recent body of work has developed neural approaches to program synthesis. Terpret (Gaunt et al., 2016) and Neural Forth (Riedel et al., 2016) use neural learning over a set of user-provided examples to complete a user-provided sketch. In neuro-symbolic synthesis (Parisotto et al., 2016) and Robust-Fill (Devlin et al., 2017), a neural architecture is used to encode a set of input-output examples and decode the resulting representation into a Flashfill program. DeepCoder (Balog et al., 2017) uses neural techniques to speed up the synthesis of Flashfill (Gulwani et al., 2015) programs.

These efforts differ from ours in goals as well as methods. Our problem is simpler, as it is conditioned on syntactic, rather than semantic, facets of programs. This allows us to generate programs in a complex programming language over a large number of data types and API methods, without needing a human-provided sketch. The key methodological difference between our work and symbolic program synthesis lies in our use of data, which allows us to generalize from a very small amount of specification. Unlike our approach, most neural approaches to program synthesis do not combine learning and combinatorial techniques. The prominent exception is Deepcoder (Balog et al., 2017), whose relationship with our work was discussed in Section 1.

## 8 CONCLUSION

We have given a method for generating type-safe programs in a Java-like language, given a label containing a small amount of information about a program's code or metadata. Our main idea is to learn a model that can predict *sketches* of programs relevant to a label. The predicted sketches are concretized into code using combinatorial techniques. We have implemented our ideas in BAYOU, a system for the generation of API-heavy code. Our experiments indicate that the system can often generate complex method bodies from just a few tokens, and that learning at the level of sketches is key to performing such generation effectively.

An important distinction between our work and classical program synthesis is that our generator is conditioned on uncertain, syntactic information about the target program, as opposed to hard constraints on the program's semantics. Of course, the programs that we generate are type-safe, and therefore guaranteed to satisfy certain semantic constraints. However, these constraints are invariant across generation tasks; in contrast, traditional program synthesis permits instance-specific semantic constraints. Future work will seek to condition program generation on syntactic labels *as well as* semantic constraints. As mentioned earlier, learning correlations between the syntax and semantics of programs written in complex languages is difficult. However, the approach of first generating and then concretizing a sketch could reduce this difficulty: sketches could be generated using a *limited amount of* semantic information, and the concretizer could use logic-based techniques (Alur et al., 2013; Gulwani et al., 2017) to ensure that the programs synthesized from these sketches match the semantic constraints exactly. A key challenge here would be to calibrate the amount of semantic information on which sketch generation is conditioned.

**Acknowledgements**   This research was supported by DARPA MUSE award #FA8750-14-2-0270 and a Google Research Award.

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

```
String s;                                          String s;
BufferedReader br;                                 BufferedReader br;
FileReader fr;                                      FileReader fr;
try {                                               try {
 fr = new FileReader($String);                       fr = new FileReader($File);
 br = new BufferedReader(fr);                         br = new BufferedReader(fr);
 while ((s = br.readLine()) != null) {}              while ((s = br.readLine()) != null){}
 br.close();                                          br.close();
} catch (FileNotFoundException _e) {                } catch (FileNotFoundException _e){
  _e.printStackTrace();                             } catch (IOException _e){
} catch (IOException _e) {                           }
  _e.printStackTrace();
}
```

(a)                                                (b)

Figure 7: Programs generated in a typical run of BAYOU, given the API method name `readLine` and the type `FileReader`.

## A  THE AML LANGUAGE

Prog   ::=   **skip** | Prog$_1$; Prog$_2$ | **call** Call |
             **let** $x$ = Call |
             **if** Exp **then** Prog$_1$ **else** Prog$_2$ |
             **while** Exp **do** Prog$_1$ | **try** Prog$_1$ Catch
Exp    ::=   Sexp | Call | **let** $x$ = Call : Exp$_1$
Sexp   ::=   $c$ | $x$
Call   ::=   Sexp$_0$.$a$(Sexp$_1$, ..., Sexp$_k$)
Catch  ::=   **catch**($x_1$) Prog$_1$ ... **catch**($x_k$) Prog$_k$

Figure 8: Grammar for AML

AML is a core language that is designed to capture the essence of API usage in Java-like languages. Now we present this language.

AML uses a finite set of *API data types*. A type is identified with a finite set of *API method names* (including constructors); the type for which this set is empty is said to be *void*. Each method name $a$ is associated with a *type signature* $(\tau_1, \ldots, \tau_k) \rightarrow \tau_0$, where $\tau_1, \ldots, \tau_k$ are the method's input types and $\tau_0$ is its return type. A method for which $\tau_0$ is void is interpreted to not return a value.

Finally, we assume predefined universes of constants and variable names.

The grammar for AML is as in Figure 8. Here, $x, x_1, \ldots$ are variable names, $c$ is a constant, and $a$ is a method name. The syntax for programs Prog includes method calls, loops, branches, statement sequencing, and exception handling. We use variables to feed the output of one method into another, and the keyword **let** to store the return value of a call in a fresh variable. Exp stands for (object-valued) expressions, which include constants, variables, method calls, and let-expressions such as "**let** $x$ = Call : Exp", which stores the return value of a call in a fresh variable $x$, then uses this binding to evaluate the expression Exp. (Arithmetic and relational operators are assumed to be encompassed by API methods.)

The operational semantics and type system for AML are standard, and consequently, we do not describe these in detail.

## B  ABSTRACTING AML PROGRAMS INTO SKETCHES

We define the abstraction function $\alpha$ for the AML language in Figure 9.

## C  NEURAL NETWORK DETAILS

In this section we present the details of the neural networks used by BAYOU.

### C.1  THE ENCODER

The task of the neural encoder is to implement the encoding function $f$ for labels, which accepts an element from a label, say $\mathsf{X}_{Calls,i}$ as input and maps it into a vector in $d$-dimensional space, where $d$ is the dimensionality of the latent space of $Z$. To achieve this, we first convert each element $\mathsf{X}_{Calls,i}$ into its one-hot vector representation, denoted $\mathsf{X}'_{Calls,i}$. Then, let $h$ be the number of neural hidden

$$
\begin{aligned}
\alpha(\textbf{skip}) &= \textbf{skip} \\
\alpha(\textbf{call } \mathsf{Sexp}_0.a(\mathsf{Sexp}_1,\ldots,\mathsf{Sexp}_k)) &= \textbf{call } \tau_0.a(\tau_1,\ldots,\tau_k) \text{ where } \tau_i \text{ is the type of } \mathsf{Sexp}_i \\
\alpha(\mathsf{Prog}_1;\mathsf{Prog}_2) &= \alpha(\mathsf{Prog}_1); \alpha(\mathsf{Prog}_2) \\
\alpha(\textbf{let } x = \mathsf{Sexp}_0.a(\mathsf{Sexp}_1,\ldots,\mathsf{Sexp}_k)) &= \textbf{call } \tau_0.a(\tau_1,\ldots,\tau_k) \text{ where } \tau_i \text{ is the type of } \mathsf{Sexp}_i \\
\alpha(\textbf{if } \mathsf{Exp} \textbf{ then } \mathsf{Prog}_1 \textbf{ else } \mathsf{Prog}_2) &= \textbf{if } \alpha(\mathsf{Exp}) \textbf{ then } \alpha(\mathsf{Prog}_1) \textbf{ else } \alpha(\mathsf{Prog}_2) \\
\alpha(\textbf{while } \mathsf{Exp} \textbf{ do } \mathsf{Prog}) &= \textbf{while } \alpha(\mathsf{Cond}) \textbf{ do } \alpha(\mathsf{Prog}) \\
\alpha(\textbf{try } \mathsf{Prog} \textbf{ catch}(x_1) \mathsf{Prog}_1 \ldots \textbf{ catch}(x_k) \mathsf{Prog}_k) &= \textbf{try } \alpha(\mathsf{Prog}) \\
&\qquad \textbf{catch}(\tau_1)\,\alpha(\mathsf{Prog}_1)\ldots \textbf{catch}(\tau_k)\,\alpha(\mathsf{Prog}_k) \\
&\qquad \text{where } \tau_i \text{ is the type of } x_i \\
\alpha(\mathsf{Exp}) &= [\,] \text{ if } \mathsf{Exp} \text{ is a constant or variable name} \\
\alpha(\mathsf{Sexp}_0.a(\mathsf{Sexp}_1,\ldots,\mathsf{Sexp}_k)) &= [\tau_0.a(\tau_1,\ldots,\tau_k)] \text{ where } \tau_i \text{ is the type of } \mathsf{Sexp}_i \\
\alpha(\textbf{let } x = \mathsf{Call}:\mathsf{Exp}_1) &= append(\alpha(\mathsf{Call}), \alpha(\mathsf{Exp}_1))
\end{aligned}
$$

Figure 9: The abstraction function $\alpha$.

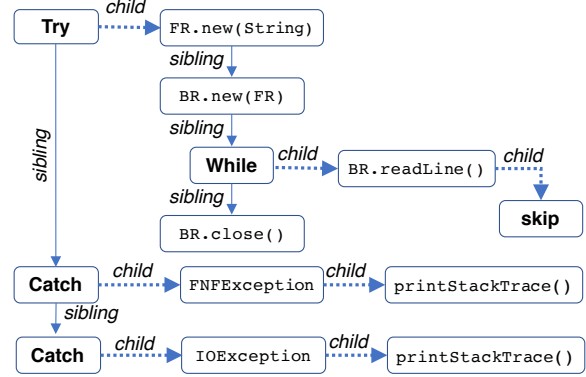

Figure 10: Tree representation of the sketch in Figure 7(a)

units in the encoder for API calls, and let $\mathbf{W}_h \in \mathbb{R}^{|Calls| \times h}$, $\mathbf{b}_h \in \mathbb{R}^h$, $\mathbf{W}_d \in \mathbb{R}^{h \times d}$, $\mathbf{b}_d \in \mathbb{R}^d$ be real-valued weight and bias matrices of the neural network. The encoding function $f(\mathsf{X}_{Calls,i})$ can be defined as follows:

$$
f(\mathsf{X}_{Calls,i}) = \tanh((\mathbf{W}_h \, . \, \mathsf{X}'_{Calls,i} + b_h) \, . \, \mathbf{W}_d + b_d)
$$

where tanh is a non-linearity defined as $\tanh(x) = \frac{1 - e^{-2x}}{1 + e^{-2x}}$. This would map any given API call into a $d$-dimensional real-valued vector. The values of entries in the matrices $\mathbf{W}_h$, $b_h$, $\mathbf{W}_d$ and $b_d$ will be learned during training. The encoder for types can be defined analogously, with its own set of matrices and hidden state.

## C.2 THE DECODER

The task of the neural decoder is to implement the sampler for $\mathsf{Y} \sim P(Y|\mathsf{Z})$. This is implemented recursively via repeated samples of production rules $\mathsf{Y}_i$ in the grammar of sketches, drawn as $\mathsf{Y}_i \sim P(Y_i|\mathbf{Y}_{i-1}, \mathsf{Z})$, where $\mathbf{Y}_{i-1} = \mathsf{Y}_1, \ldots, \mathsf{Y}_{i-1}$. The generation of each $\mathsf{Y}_i$ requires the generation of a new "path" from a series of previous "paths", where each path corresponds to a series of production rules fired in the grammar.

As a sketch is tree-structured, we use a top-down tree-structured recurrent neural network similar to Zhang et al. (2016), which we elaborate in this section. First, similar to the notion of a "dependency path" in Zhang et al. (2016), we define a *production path* as a sequence of pairs $\langle (v_1, e_1), (v_2, e_2), \ldots, (v_k, e_k) \rangle$ where $v_i$ is a node in the sketch (i.e., a term in the grammar) and $e_i$ is the type of edge that connects $v_i$ with $v_{i+1}$. Our representation has two types of edges: *sibling* and *child*. A sibling edge connects two nodes at the same level of the tree and under the same parent node (i.e., two terms in the RHS of the same rule). A child edge connects a node with another that is

$$h_0 = \mathbf{W}_l.\mathsf{Z} + \mathbf{b}_l \qquad\qquad y_i = \left\{ \begin{array}{ll} \mathsf{softmax}(\mathbf{W}_y^c . h_i + \mathbf{b}_y^c) & \text{if } e_i = child \\ \mathsf{softmax}(\mathbf{W}_y^s . h_i + \mathbf{b}_y^s) & \text{if } e_i = sibling \end{array} \right.$$

$$\tag{2}$$

$$h_i^c = \mathbf{W}_h^c . h_{i-1} + \mathbf{b}_h^c + \mathbf{W}_v^c . v_i' + \mathbf{b}_v^c \qquad \text{where}$$

$$h_i^s = \mathbf{W}_h^s . h_{i-1} + \mathbf{b}_h^s + \mathbf{W}_v^s . v_i' + \mathbf{b}_v^s \qquad \tanh(x) = \frac{1 - e^{-2x}}{1 + e^{-2x}} \quad \text{and}$$

$$h_i = \left\{ \begin{array}{ll} \tanh(h_i^c) & \text{if } e_i = child \\ \tanh(h_i^s) & \text{if } e_i = sibling \end{array} \right. \qquad \mathsf{softmax}(\mathbf{x})_j = \frac{e^{x_j}}{\sum_{k=1}^{K} e^{x_k}} \quad \text{for } j \in 1 \ldots K$$

Figure 11: Computing the hidden state and output of the decoder

one level deeper in the tree (i.e., the LHS with a term in the RHS of a rule). We consider a sequence of API calls connected by sequential composition as siblings. The root of the entire tree is a special node named root, and so the first pair in all production paths is $(\mathsf{root}, child)$. The last edge in a production path is irrelevant $(\cdot)$ as it does not connect the node to any subsequent nodes.

As an example, consider the sketch in Figure 7(a), whose representation as a tree for the decoder is shown in Figure 10. For brevity, we use $s$ and $c$ for $sibling$ and $child$ edges respectively, abbreviate some classnames with uppercase letters in their name, and omit the first pair $(\mathsf{root}, c)$ that occurs in all paths. There are four production paths in the tree of this sketch:

1. $(\mathbf{try}, c), (\texttt{FR.new(String)}, s), (\texttt{BR.new(FR)}, s), (\mathbf{while}, c), (\texttt{BR.readLine()}, c), (\mathbf{skip}, \cdot)$
2. $(\mathbf{try}, c), (\texttt{FR.new(String)}, s), (\texttt{BR.new(FR)}, s), (\mathbf{while}, s), (\texttt{BR.close()}, \cdot)$
3. $(\mathbf{try}, s), (\mathbf{catch}, c), (\texttt{FNFException}, c), (\texttt{T.printStackTrace()}, \cdot)$
4. $(\mathbf{try}, s), (\mathbf{catch}, s), (\mathbf{catch}, c), (\texttt{IOException}, c), (\texttt{T.printStackTrace()}, \cdot)$

Now, given a $\mathsf{Z}$ and a sequence of pairs $\mathbf{Y}_i = \langle (v_1, e_1), \ldots, (v_i, e_i) \rangle$ along a production path, the next node in the path is assumed to be dependent solely on $\mathsf{Z}$ and $\mathbf{Y}_i$. Therefore, a single inference step of the decoder computes the probability $P(v_{i+1}|\mathbf{Y}_i, \mathsf{Z})$. To do this, the decoder uses two RNNs, one for each type of edge $c$ and $s$, that act on the production pairs in $\mathbf{Y}_i$. First, all nodes $v_i$ are converted into their one-hot vector encoding, denoted $v_i'$.

Let $h$ be the number of hidden units in the decoder, and $|G|$ be the size of the decoder's output vocabulary, i.e., the total number of terminals and non-terminals in the grammar of sketches. Let $\mathbf{W}_h^e \in \mathbb{R}^{h \times h}$ and $\mathbf{b}_h^e \in \mathbb{R}^d$ be the decoder's hidden state weight and bias matrices, $\mathbf{W}_v^e \in \mathbb{R}^{|G| \times h}$ and $\mathbf{b}_v^e \in \mathbb{R}^h$ be the input weight and bias matrices, and $\mathbf{W}_y^e \in \mathbb{R}^{h \times |G|}$ and $\mathbf{b}_y^e \in \mathbb{R}^{|G|}$ be the output weight and bias matrices, where $e$ is the type of edge: either $c$ (child) or $s$ (sibling). We also use "lifting" matrices $\mathbf{W}_l \in \mathbb{R}^{d \times h}$ and $\mathbf{b}_l \in \mathbb{R}^h$, to lift the $d$-dimensional vector $\mathsf{Z}$ onto the (typically) higher-dimensional hidden state space $h$ of the decoder.

Let $h_i$ and $y_i$ be the hidden state and output of the network at time point $i$. We compute these quantities as given in Figure 11, where $\tanh$ is a non-linear activation function that converts any given value to a value between -1 and 1, and $\mathsf{softmax}$ converts a given $K$-sized vector of arbitrary values to another $K$-sized vector of values in the range $[0, 1]$ that sum to 1—essentially a probability distribution.

The type of edge at time $i$ decides which RNN to choose to update the (shared) hidden state $h_i$ and the output $y_i$. Training consists of learning values for the entries in all the $\mathbf{W}$ and $\mathbf{b}$ matrices. During training, $v_i'$, $e_i$ and the target output are known from the data point, and so we optimize a standard cross-entropy loss function (over all $i$) between the output $y_i$ and the target output. During inference, $P(v_{i+1}|\mathbf{Y}_i, \mathsf{Z})$ is simply the probability distribution $y_i$, the result of the $\mathsf{softmax}$.

A sketch is obtained by starting with the root node pair $(v_1, e_1) = (\mathsf{root}, child)$, recursively applying Equation 2 to get the output distribution $y_i$, sampling a value for $v_{i+1}$ from $y_i$, and growing the tree by adding the sampled node to it. The edge $e_{i+1}$ is provided as $c$ or $s$ depending on the $v_{i+1}$ that was

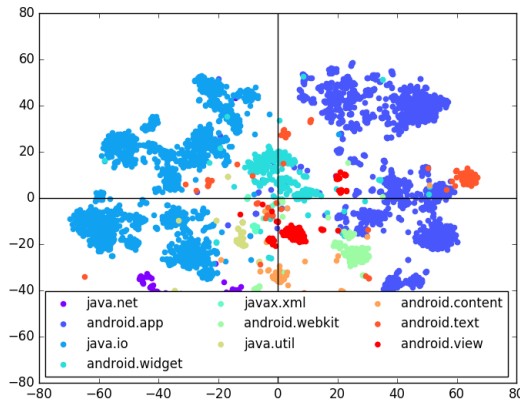

Figure 12: 2-dimensional projection of latent space of the GSNN-Sk model

sampled. If only one type of edge is feasible (for instance, if the node is a terminal in the grammar, only a *sibling* edge is possible with the next node), then only that edge is provided. If both edges are feasible, then both possibilities are recursively explored, growing the tree in both directions.

**Remarks.** In our implementation, we generate trees in a depth-first fashion, by exploring a *child* edge before a *sibling* edge if both are possible. If a node has two children, a neural encoding of the nodes that were generated on the left is carried onto the right sub-tree so that the generation of this tree can leverage additional information about its previously generated sibling. We refer the reader to Section 2.4 of Zhang et al. (2016) for more details.

## D    ADDITIONAL EVALUATION

In this section, we provide results of additional experimental evaluation.

### D.1    CLUSTERING

Similar to the visualization of the 2-dimensional latent space in Figure 5, we also plotted the latent space of the GSNN-Sk model trained on sketches. Figure 12 shows this plot. We observed that the latent space is clustered, relatively, more densely than that of our model (keep in mind that the plot colors are different when comparing them).

### D.2    QUALITATIVE EVALUATION

To give a sense of the quality of the end-to-end generation, we present and discuss a few usage scenarios for our system, BAYOU. In each scenario, we started with a set of API calls, types or keywords as labels that indicate what we (as the user) would like the generated code to perform. We then pick a single program in the top-5 results returned by BAYOU and discuss it. Figure 13 shows three such example usage scenarios.

In the first scenario, we would like the system to generate a program to write something to a file by calling `write` using the type `FileWriter`. With this label, we invoked BAYOU and it returned with a program that actually accomplishes the task. Note that even though we only specified `FileWriter`, the program uses it to feed a `BufferedWriter` to write to a file. This is an interesting pattern learned from data, that file reads and writes in Java often take place in a buffered manner. Also note that the program correctly flushes the buffer before closing it, even though none of this was explicitly specified in the input.

In the second scenario, we would like the generated program to set the title and message of an Android dialog. This time we provide no API calls or types but only keywords. With this, BAYOU generated a program that first builds an Android dialog box using the helper class `AlertDialog.Builder`, and does set its title and message. In addition, the program also adds a

| Input to BAYOU | Generated program ranked among the top-5 |
|---|---|
| $X_{Types} = \{\texttt{FileWriter}\}$
$X_{Calls} = \{\texttt{write}\}$
$X_{Keys} = \emptyset$ | ```java
BufferedWriter bw;
FileWriter fw;
try {
    fw = new FileWriter($String, $boolean);
    bw = new BufferedWriter(fw);
    bw.write($String);
    bw.newLine();
    bw.flush();
    bw.close();
} catch (IOException _e) {
}
``` |
| $X_{Types} = \emptyset$
$X_{Calls} = \emptyset$
$X_{Keys} = \{\text{android, dialog, set,}$
$\quad\text{title, message}\}$ | ```java
Builder builder2;
Builder builder1;
AlertDialog alertDialog;
Builder builder4;
Builder builder3;
builder1 = new Builder($Context);
builder2 = builder1.setTitle($String);
builder3 = builder2.setMessage($String);
builder4 = builder3.setNeutralButton($String,
                            $OnClickListener);
alertDialog = builder4.show();
``` |
| $X_{Types} = \emptyset$
$X_{Calls} = \{\texttt{startPreview}\}$
$X_{Keys} = \emptyset$ | ```java
Parameters parameters;
parameters = $Camera.getParameters();
parameters.setPreviewSize($int, $int);
parameters.setRecordingHint($boolean);
$Camera.setParameters(parameters);
$Camera.startPreview();
``` |

Figure 13: Qualitative usage scenarios of BAYOU.

button to the dialog box – another interesting pattern learned from data that dialog boxes in Android often have a button, typically to close the dialog. Finally it shows the dialog with these items.

In the final scenario, we would like BAYOU to generate code to start preview mode in the phone's camera. We provided simply the API call `startPreview` as input. With this, the system was automatically able to recognize that we are interested in the camera API, and generate a program that accomplishes the task. Note that the program first obtains the camera parameters, and sets the preview display size (the `int` arguments are the width and height) before starting the preview. We confirmed from the Android `Camera` API documentation that this is recommended practice, and the model appears to have learned this automatically from data.

