# OpenReview forum: "Neural Sketch Learning for Conditional Program Generation"
_ICLR.cc/2018/Conference — Accept (Oral)_

### Official Review · AnonReviewer3 · 2017-11-26
**sketch learning**

**Rating:** 7
**Confidence:** 2

**Review:**

The authors introduce an algorithm in the subfield of conditional program generation that is able to create programs in a rich java like programming language. In this setting, they propose an algorithm based on sketches- abstractions of programs that capture the structure but discard program specific information that is not generalizable such as variable names. Conditioned on information such as type specification or keywords of a method they generate the method's body from the trained sketches.

Positives:

	•	Novel algorithm and addition of rich java like language in subfield of 'conditional program generation' proposed
	•	Very good abstract: It explains high level overview of topic and sets it into context plus gives a sketch of the algorithm and presents the positive results.
	•	Excellently structured and presented paper

	•	Motivation given in form of relevant applications and mention that it is relatively unstudied
	•	The hypothesis/ the papers goal is clearly stated. It is introduced with 'We ask' followed by two well formulated lines that make up the hypothesis. It is repeated multiple times throughout the paper. Every mention introduces either a new argument on why this is necessary or sets it in contrast to other learners, clearly stating discrepancies.
	•	Explanations are exceptionally well done: terms that might not be familiar to the reader are explained. This is true for mathematical aspects as well as program generating specific terms. Examples are given where appropriate in a clear and coherent manner
	•	Problem statement well defined mathematically and understandable for a broad audience
	•	Mentioning of failures and limitations demonstrates a realistic  view on the project
	•	Complexity and time analysis provided
	•	Paper written so that it's easy for a reader to implement the methods
	•	Detailed descriptions of all instantiations even parameters and comparison methods
	•	System specified
	•	Validation method specified
	•	Data and repository, as well as cleaning process provided
	•	Every figure and plot is well explained and interpreted
	•	Large successful evaluation section provided
	•	Many different evaluation measures defined to measure different properties of the project
	•	Different observability modes
	•	Evaluation against most compatible methods from other sources
	•	 Results are in line with hypothesis
	•	Thorough appendix clearing any open questions

It would have been good to have a summary/conclusion/future work section

SUMMARY: ACCEPT.  The authors present a very intriguing novel approach that  in a clear and coherent way. The approach is thoroughly explained for a large audience. The task itself is interesting and novel. The large evaluation section that discusses many different properties is a further indication that this approach is not only novel but also very promising. Even though no conclusive section is provided, the paper is not missing any information.

---

> ### Author Response · Authors · 2017-12-08
> **Thank you**
>
> Thank you for your feedback about the paper. We will add a conclusion section to the final version of the paper.

---

### Official Review · AnonReviewer1 · 2017-11-27
**Well executed, moving towards more realistic program synthesis tasks.**

**Rating:** 8
**Confidence:** 4

**Review:**

This paper aims to synthesize programs in a Java-like language from a task description (X) that includes some names and types of the components that should be used in the program. The paper argues that it is too difficult to map directly from the description to a full program, so it instead formulates the synthesis in two parts. First, the description is mapped to a "sketch" (Y) containing high level program structure but no concrete details about, e.g., variable names. Afterwards, the sketch is converted into a full program (Prog) by stochastically filling in the abstract parts of the sketch with concrete instantiations.

The paper presents an abstraction method for converting a program into a sketch, a stochastic encoder-decoder model for converting descriptions to trees, and rejection sampling-like approach for converting sketches to programs. Experimentally, it is shown that using sketches as an intermediate abstraction outperforms directly mapping to the program AST. The data is derived from an online repository of ~1500 Android apps, and from that were extracted ~150k methods, which makes the data very respectable in terms of realisticness and scale. This is one of the strongest points of the paper.

One point I found confusing is how exactly the Combinatorial Concretization step works. Am I correct in understanding that this step depends only on Y, and that given Y, Prog is conditionally independent of X? If this is correct, how many Progs are consistent with a typical Y? Some additional discussion of why no learning is required for the P(Prog | Y) step would be appreciated.

I'm also curious whether using a stochastic latent variable (Z) is necessary. Would the approach work as well using a more standard encoder-decoder model with determinstic Z?

Some discussion of Grammar Variational Autoencoder (Kusner et al) would probably be appropriate.

Overall, I really like the fact that this paper is aiming to do program synthesis on programs that are more like those found "in the wild". While the general pattern of mapping a specification to abstraction with a neural net and then mapping the abstraction to a full program with a combinatorial technique is not necessarily novel, I think this paper adds an interesting new take on the pattern (it has a very different abstraction than say, DeepCoder), and this paper is one of the more interesting recent papers on program synthesis using machine learning techniques, in my opinion.

---

> ### Author Response · Authors · 2017-12-08
> **Answering questions in the feedback**
>
> Thank you for your feedback about the paper. We answer your specific questions below.
>
> Question: Am I correct in understanding that [Combinatorial Concretization] step depends only on Y, and that given Y, Prog is conditionally independent of X? If this is correct, how many Progs are consistent with a typical Y?
>
> Answer: Yes, Prog is conditionally independent of X given a sketch Y. In theory, there may be an infinite number of Progs for every Y. A simple example is two Progs that differ only in variable names, thereby corresponding to the same Y; for another example, there can be very many expressions that match the type of an API method argument. However, in practice, we use certain heuristics to limit the space of Progs from a given Y (these heuristics are abstractly captured by the distribution P(Prog | Y). In particular, these heuristics prioritize smaller, simpler programs over complex ones, and name local variables in a canonical way.
>
> While we didn't collect this data systematically, our experience with the system suggests that under the heuristics actually implemented in it, a typical Y leads to only ~5-10 distinct Progs in our experiments. We will collect this data more thoroughly and add it to the paper.
>
> Question: Some additional discussion of why no learning is required for the P(Prog | Y) step would be appreciated.
>
> Answer: In principle, this step could be made data-driven; however, the resulting learning problem would be very difficult. This is because a single sketch used for training can correspond to many training programs that only differ in superficial details (for example local variable names). Learning to decide which differences between programs are superficial and which are not, solely by looking at the syntax of programs, is hard. In contrast, our approach of heuristically choosing P(Prog | Y) utilizes our domain knowledge of language semantics (for example, that local variable names do not matter, and that some algebraic expressions are semantically equivalent). This knowledge allows us to limit the set of programs that we end up generating. We will clarify this in more detail in the paper.
>
>
> Question: I'm also curious whether using a stochastic latent variable (Z) is necessary. Would the approach work as well using a more standard encoder-decoder model with deterministic Z?
>
> Answer: The randomness associated with the latent variable Z serves as a way to regularize the learning process (a similar argument is made in the context of VAEs for the stochastic latent variable used during VAE learning). We were concerned that without the stochasticity (i.e., with a deterministic Z), training the model would be more likely to be affected by overfitting. Practically speaking, the stochasticity also serves as a way to ensure that we can generate a wide variety of possible programs from a given X. If Z was not random, a particular set of labels X will always result in exactly the same value of Z.
>
> Comment: Some discussion of Grammar Variational Autoencoder (Kusner et al) would probably be appropriate.
>
> Answer: Kusner et al’s work proposes a VAE for context-free grammars. Being an auto-encoder it is a generative model, but it is not a conditional model such as ours. In their application towards synthesizing molecular structures, given a particular molecular structure, their model can be used to search the latent space for similar valid structures. In our setting, however, we are not given a sketch but only labels about the sketch, and our task is learn a conditional model that can predict a whole sketch given labels.
>
> We will add the discussion about this work in the final version of the paper.

---

### Official Review · AnonReviewer2 · 2017-12-02
**Sketch Learning for Program Generation**

**Rating:** 7
**Confidence:** 3

**Review:**

This is a very well-written and nicely structured paper that tackles the problem of generating/inferring code given an incomplete description (sketch) of the task to be achieved. This is a novel contribution to existing machine learning approaches to automated programming that is achieved by training on a large corpus of Android apps. The combination of the proposed technique and leveraging of real data are a substantial strength of the work compared to many approaches that have come previously.

This paper has many strengths:
1) The writing is clear, and the paper is well-motivated
2) The proposed algorithm is described in excellent detail, which is essential to reproducibility
3) As stated previously, the approach is validated with a large number of real Android projects
4) The fact that the language generated is non-trivial (Java-like) is a substantial plus
5) Good discussion of limitations

Overall, this paper is a valuable addition to the empirical software engineering community, and a nice break from more traditional approaches of learning abstract syntax trees.

---

> ### Author Response · Authors · 2017-12-08
> **Thank you**
>
> Thank you for your feedback about the paper.

---

### Author Response · Authors · 2018-01-05
**Summary of changes in the latest version**

We have uploaded the final version of the paper making the following changes:
1. Clarification about learning the distribution P(Prog | Y).
2. Discussion about the related work on Grammar VAE (Kusner et al).
3. Addition of a conclusion section to the paper.

---

### Decision · Program_Chairs · 2018-01-29
**ICLR 2018 Conference Acceptance Decision**

**Decision:**

Accept (Oral)

**Comment:**

This paper presents a novel and interesting sketch-based approach to conditional program generation. I will say upfront that it is worth of acceptance, based on its contribution and the positivity of the reviews. I am annoyed to see that the review process has not called out the authors' lack of references to the decently body of existing work on generating structure on neural sketch programming and on generating under grammatical constraint. The authors' will need look no further than the proceedings of the *ACL conferences of the last few years to find papers such as:
* Dyer, Chris, et al. "Recurrent Neural Network Grammars." Proceedings of NAACL-HLT (2016).
* Kuncoro, Adhiguna, et al. "What Do Recurrent Neural Network Grammars Learn About Syntax?." Proceedings of EACL (2016).
* Yin, Pengcheng, and Graham Neubig. "A Syntactic Neural Model for General-Purpose Code Generation." Proceedings of ACL (2017).
* Rabinovich, Maxim, Mitchell Stern, and Dan Klein. "Abstract Syntax Networks for Code Generation and Semantic Parsing." Proceedings of ACL (2017).

Or other work on neural program synthesis, with sketch based methods:
* Gaunt, Alexander L., et al. "Terpret: A probabilistic programming language for program induction." arXiv preprint arXiv:1608.04428 (2016).
* Riedel, Sebastian, Matko Bosnjak, and Tim Rocktäschel. "Programming with a differentiable forth interpreter." CoRR, abs/1605.06640 (2016).

Likewise the references to the non-neural program synthesis and induction literature are thin, and the work is poorly situated as a result.

It is a disappointing but mild failure of the scientific process underlying peer review for this conference that such comments were not made. The authors are encouraged to take heed of these comments in preparing their final revision, but I will not object to the acceptance of the paper on these grounds, as the methods proposed therein are truly interesting and exciting.